# Challenges in Diagnosing COVID-19-Associated Pulmonary Aspergillosis in Critically Ill Patients: The Relationship between Case Definitions and Autoptic Data

**DOI:** 10.3390/jof8090894

**Published:** 2022-08-23

**Authors:** Giacomo Casalini, Andrea Giacomelli, Laura Galimberti, Riccardo Colombo, Elisabetta Ballone, Giacomo Pozza, Martina Zacheo, Miriam Galimberti, Letizia Oreni, Luca Carsana, Margherita Longo, Maria Rita Gismondo, Cristina Tonello, Manuela Nebuloni, Spinello Antinori

**Affiliations:** 1Department of Biomedical and Clinical Sciences, Università degli Studi di Milano, 20157 Milan, Italy; 2III Division of Infectious Diseases, ASST Fatebenefratelli-Sacco, Luigi Sacco Hospital, 20157 Milan, Italy; 3Department of Anesthesiology and Intensive Care, ASST Fatebenefratelli-Sacco, Luigi Sacco Hospital, 20157 Milan, Italy; 4Pathology Unit, ASST Fatebenefratelli-Sacco, Luigi Sacco Hospital, 20157 Milan, Italy; 5Clinical Microbiology, Virology and Bioemergency Diagnostics, ASST Fatebenefratelli-Sacco, Luigi Sacco Hospital, 20157 Milan, Italy

**Keywords:** COVID-19, invasive fungal infections, CAPA, aspergillosis, autopsy, histopathology

## Abstract

Critically ill COVID-19 patients can develop invasive pulmonary aspergillosis (CAPA). Considering the weaknesses of diagnostic tests/case definitions, as well as the results from autoptic studies, there is a debate on the real burden of aspergillosis in COVID-19 patients. We performed a retrospective observational study on mechanically ventilated critically ill COVID-19 patients in an intensive care unit (ICU). The primary objective was to determine the burden of CAPA by comparing clinical diagnosis (through case definitions/diagnostic algorithms) with autopsy results. Twenty patients out of 168 (11.9%) developed probable CAPA. Seven (35%) were females, and the median age was 66 [IQR 59–72] years. Thirteen CAPA patients (65%) died and, for six, an autopsy was performed providing a proven diagnosis in four cases. Histopathology findings suggest a focal pattern, rather than invasive and diffuse fungal disease, in the context of prominent viral pneumonia. In a cohort of mechanically ventilated patients with probable CAPA, by performing a high rate of complete autopsies, invasive aspergillosis was not always proven. It is still not clear whether aspergillosis is the major driver of mortality in patients with CAPA.

## 1. Introduction

Critically ill COVID-19 patients admitted to an intensive care unit (ICU) are susceptible to secondary infections which negatively affect outcome [1]. Since the beginning of the pandemic, invasive pulmonary aspergillosis (IPA) emerged as a potential secondary infection in severe COVID-19 patients, resembling the well-known association between influenza-related acute respiratory distress syndrome (ARDS) and IPA [2]. The main hypothesis about COVID-19-associated pulmonary aspergillosis (CAPA) pathogenesis is that *Aspergillus* spp. invasion is promoted by alterations of the barrier function of the airway’s epithelium and by the marked dysregulation of the immune system, which is observed in patients with severe COVID-19 (the host predisposing factor) [3,4,5,6,7,8,9]. CAPA is observed almost exclusively in mechanically ventilated patients [10]. Corticosteroids use is associated with CAPA in many observational studies [11,12], even if others have opposite results [13,14]. Anti-interleukin-6 receptor use seems associated with CAPA [15]. The average incidence in mechanically ventilated patients is 10–15%, even if in earlier studies the rate was higher and sometimes over 30%, and mortality can exceed 50% [10,11,12,13,14,15,16].

Clinical diagnosis of CAPA usually relies on the performance of fungal diagnostic tests that are variably combined in diagnostic algorithms/case definitions. The performance of diagnostic tests is suboptimal and diagnostic algorithms showed many limitations. Furthermore, CAPA has non-specific radiological and clinical features, making diagnosis and management a clinical challenge [17]. A definite diagnosis of pulmonary aspergillosis is achieved with histopathological examination of lung tissue. This approach is frequently not feasible in COVID-19 patients, because critical illness and mechanical ventilation hamper invasive diagnostic procedures. To further complicate this scenario, autopsy studies which evaluate CAPA incidence question the real burden of the disease, reporting a much lower prevalence than observational studies [18,19].

We performed a retrospective observational study on critically ill patients admitted to our ICU, focusing on the relationship between the clinical diagnosis of CAPA, using different diagnostic algorithms/case definitions, and post-mortem histopathological examination of lung tissue, speculating on the real burden of invasive aspergillosis in severe COVID-19. We also describe clinical feature of patients with CAPA, reporting the incidence and outcome analysis.

## 2. Materials and Methods

*Study design, setting and participants*. We conducted a retrospective, single-centre observational study in the general ICU of the Luigi Sacco hospital in Milan, Italy. We included all adult patients with a SARS-CoV-2-positive RT-PCR assay performed with nasopharyngeal swabs consecutively admitted to ICU because of severe respiratory failure secondary to COVID-19 pneumonia, requiring orotracheal intubation and mechanical ventilation. The study was conducted over a period of 10 months (August 2020, May 2021). Patient admitted before this timeframe were not included in our analysis because a systematic screening for pulmonary aspergillosis was not introduced yet. Hospital characteristics and the internal reorganization required to deal with the pandemic have been previously described [20]. During the study period there were two consecutive pandemic waves whose characteristics have been previously described; of note, the risk of death during the epidemic waves did not change significantly [21]. The study was approved by our Comitato Etico Interaziendale Area 1, Milan, Italy (Protocol No. 16088).

*Data collection and procedures.* Anonymized patient data were securely recorded in a dedicated electronic database, from hospital admission to hospital discharge or in-hospital death. For each patient, epidemiological (age, sex, comorbidities) and clinical data (length of ICU stay, sequential organ failure assessment score—SOFA, chest imaging, anti-COVID-19 therapies, antibiotic therapy, outcome) were retrieved. COVID-19 patients admitted to ICU were systematically screened weekly with a serum galactomannan enzyme immunoassay (GM-EIA) and bronchial aspirate (BAS) culture. Bronchoalveolar lavage (BAL) was performed when clinically indicated and samples were sent for direct microscopic examination, bacterial and fungal culture, and GM-EIA. We also recorded data about secondary pulmonary bacterial infections (ventilator-associated pneumonia, VAP, or ventilator-associated tracheobronchitis, VAT) concomitant to CAPA diagnosis. Clinical and microbiological data were reviewed by an expert infectious disease (ID) physician who established the diagnosis of CAPA or ventilator-associated lower respiratory tract infections (VA-LRTI), either VAP or VAT.

*Laboratory diagnosis.* GM-EIA was performed on serum or BAL using Platelia Aspergillus galactomannan assay according to the manufacturer’s instructions (Bio-Rad, Boulevard Raymond Poincaré 92430, Marnes-la-Coquette, France). The optical density index (ODI) cut-off for a positive result was 0.5 on serum samples and 1.0 in BAL, according to the ECMM/ISHAM consensus criteria [22].

*Autopsy study.* Autopsies were performed as previously described [23]. Briefly, samples from lungs, liver, kidney, heart, bowel, brain, skin, and spleen were taken for histological analysis. A median of 7 tissue blocks (range 5–9) were taken from each lung, selecting the most representative areas of each lobe identified at macroscopic examination; additional blocks were performed if necessary. Tissues were fixed in 10% buffered formalin for >48 h. Three-µm paraffin sections were stained by haematoxylin-eosin. Methenamine silver staining (Grocott-Gomori) was also performed. Histological evaluation of lung tissues was performed in a blind manner and independently by two pathologists: histological features of the cellular and interstitial damage were described and graded by using a semiquantitative scale for percentage of tissue involved: 0 = absent, <5% = rare, 5–25% = focal, 25–50% = present, 50–75% = plurifocal, >75% = diffuse. Molecular biology for SARS-CoV-2 was done on the most representative area of each case. PCR for *Aspergillus* spp. was done in the cases with morphological detection of fungal presence. For SARS-CoV2 PCR, RNA was extracted from three 5-µm paraffin sections using Quick-RNA FFPE Miniprep (Zymo Research, Irvine, CA, USA). Two target genes, the RNA-dependent RNA polymerase gene (RdRP gene, Co-Diagnostic, Salk Lake City, UT, USA) and envelope (E gene, the WHO/Charité, Berlin, Germany), were amplified and tested in all selected samples. Appropriate positive and negative tissue samples were used as controls. For *Aspergillus* spp., DNA was extracted from three 5-µm paraffin sections; molecular analysis was performed as reported by Bretagne et al. [24].

*Definitions*. For incidence and outcome analysis, CAPA was defined according to the recent ECMM/ISHAM consensus criteria [22]. Using patient-level data, each CAPA case was reclassified with available diagnostic criteria and cases definitions: *Asp*ICU criteria by Blot [25], influenza-associated pulmonary aspergillosis (IAPA) algorithms by Verwey [26] and Schauwvlieghe [2], CAPA criteria by White [11] and the modified *Asp*ICU algorithm by Gangneux [27]. VAP and VAT were defined as proposed by Rouzé and colleagues [28].

The primary objective was to determine the burden of CAPA by comparing clinical diagnosis (through case definitions/diagnostic algorithms) with results from post-mortem histopathological lung examinations. The secondary objective included evaluation of the performance of different case definitions, the description of clinical characteristics of CAPA patients, together with an incidence and outcome analysis.

*Statistical analysis*. Continuous variables are expressed as means ± standard deviations or medians and interquartile ranges (IQR). Categorical variables are expressed as percentages. Continuous variables were compared using the Student’s *t*-test or the Mann–Whitney U test, as appropriate. Categorical variables were compared using the chi-square test or the Fisher’s exact test. The crude incidence rate was estimated as the number of incident first occurrences of CAPA in intubated patients divided by the patient days at risk (days on mechanical ventilation), with a corresponding 95% confidence interval (CI), computed using a Poisson distribution. The cumulative incidence of first occurrences of CAPA over time in intubation was estimated considering death and extubation as a competing event. All the statistical tests were two-tailed and were considered significant with a *p* less than 0.05.

## 3. Results

### 3.1. Characteristics of the Study Population

A total of 167 patients with SARS-CoV-2 infection were consecutively admitted to our ICU during the study period. All of them required orotracheal intubation and mechanical ventilation on ICU admission or immediately after. More than two-thirds of patients were male (76.8%), and the median age was 66 years (IQR 59–72). The most common comorbidity was cardiovascular disease (*n* = 96, 57.1%) with 70% of patients presenting with at least one comorbidity (Table 1).

Twenty patients (11.9%) developed probable CAPA according to the ECMM/ISHAM consensus definition (Table 2), while there were neither possible nor proven CAPA diagnoses in vivo. Crude CAPA incidence was 8 cases per 1000 patient days (95% CI 5.1–12.4). The median time from ICU admission to CAPA diagnosis was 8.5 days (IQR 3.5–10). The median time from orotracheal intubation and CAPA diagnosis was 8.5 days (IQR 4–10). Clinical baseline characteristic did not differ significantly between CAPA or non-CAPA patients. In the CAPA cohort, seven patients (35%) were females, and the median age was 66 years (IQR 60–72). Ninety percent of patients received systemic glucocorticoids vs. 89.2% of patients without CAPA.

Culture of respiratory specimens revealed a mould in six patients (30%; four BAL cultures and two BAS cultures). The species identified were *A. fumigatus* (3 cases), *A. niger* (2 cases) and rare mould colonies not otherwise specified in one case. While BAL GM was positive in nearly all patient tested (18/19, 94.7%), serum GM was positive in only five cases (25%). Chest X-ray showed bilateral interstitial infiltrates in all patients, as expected in COVID-19 ARDS. None of the patients had radiological findings suggestive of pulmonary aspergillosis (i.e., cavitation, nodules, or wedge-shaped consolidations); however, a computed tomography (CT) scan was performed in only three cases. Nine patients had a concurrent secondary pulmonary bacterial infection concomitant to CAPA diagnosis, VAP in seven cases and VAT in two cases. Thirteen (65%) patients received antifungal treatment with voriconazole, only one patient was treated with isavuconazole. Voriconazole was administered at a dose of 6 mg/Kg q12h on day one, then 4 mg/Kg q12h. Isavuconazole sulfate was administered at a dose of 200 mg q8h for the first 48 h (6 doses), then 200 mg q24h. Intravenous formulation was used until the patient was unable to swallow. Patients received antifungal therapy until hospital discharge. Therapeutic drug monitoring was performed according to the indications of the infectious disease consultant. Three patients did not receive antifungal treatment because of transition to palliative care (patient #3, #12 and #15,), two patients recovered spontaneously without antifungals (patient #13 and #16) and, in one case, death occurred before treatment initiation (patient #20). The in-hospital mortality in CAPA patients was 65%.

### 3.2. Autopsy Study

Autopsy was performed in 6 out of 13 deceased CAPA patients (46.2%). All six patients received a clinical diagnosis of probable CAPA, according to the ECMM/ISHAM definition.

Post-mortem lung histopathological findings included bilateral diffuse alveolar damage (DAD) in all patients, with prominent diffuse type II pneumocyte hyperplasia and atypia, myofibroblast proliferation, plurifocal hyaline membrane formation, and foci of obliterating fibrosis (Figure 1).

Platelet-fibrin thrombi in small arterial vessels (<1 mm in diameter) were found in three cases. In four out of the six autopsy cases, the histological examination showed the presence of fungal hyphae of *Aspergillus* spp. In three cases, fungi were associated with exudative inflammatory and necrotic lung lesions (two of them were large necrotic areas and fungal hyphae were abundant; one was a small area of exudative inflammation and fungal hyphae were rare); in the last, fourth case, fungi were found in the bronchial lumen, associated with acute inflammation and necrosis of the bronchial wall. Fungal angioinvasion was found in none of the four cases (Figure 2). PCR for SARS-CoV-2 was positive in all the lung tissues tested. PCR for *Aspergillus* spp. was positive in three out of the four cases with histological diagnosis of aspergillosis; the case that was negative was the one with the small area of exudative inflammation and rare hyphae.

Among the proven CAPA cases, two patients died despite voriconazole treatment (#1 and #9), one patient was transitioned to palliative care (#12) and one patient (#19) died before treatment initiation. The cultures of respiratory specimens were negative for fungi in all patients. Serum GM was positive in two cases (ODI 0.5 and 1.7) while BAL GM was positive in all patients (ODI range 1–12.5). Interestingly, in the two patients without autoptic CAPA confirmation, BAL GM was positive (and, in patient #6, ODI decreased during treatment) and, in one case, the BAS culture revealed *A. niger* colonies, probably reflecting airway colonization. Death occurred despite antifungal treatment.

We finally reviewed all the autopsies performed on COVID-19-deceased patients in our institutions during the study period and we found two more cases of proven CAPA, so that the overall autoptic prevalence of CAPA was 4.8% (6 out of 125 autopsies). These two patients were both admitted to medical wards because of severe COVID-19 pneumonia which required high-flow oxygen therapy. Due to underlying medical comorbidities, intensive treatment was not indicated, and both patients died a few days after admission without clinical suspicion of CAPA (fungal diagnostic tests were not performed in vivo). Autoptic diagnosis was made by both direct observation of fungal hyphae and positive *Aspergillus* spp. PCR.

### 3.3. In Vivo CAPA Classification

Each patient with a clinical diagnosis of probable CAPA according to the ECMM/ISHAM consensus definition was then re-classified using different diagnostic algorithms (Table 3). We found that the most sensitive criteria were those from Verwey and Schauwvlieghe, which identified 20 probable cases. Using *Asp*ICU criteria by Blot (the first diagnostic algorithm introduced in ICU for IPA diagnosis) we could not identify any putative aspergillosis; even if patients had positive clinical and radiological criteria, as well as BAL cultures positive for *Aspergillus* spp., none of them had underlying classical host risk factors (COVID-19 was not, hence, considered a host risk factor) or a positive direct examination of respiratory specimen showing branching hyphae (even if cytological examination of BAL was always performed). However, we identified six patients with *Aspergillus* spp. airway colonization. Using the *Asp*ICU algorithm by Gangneux, we did not identify any putative CAPA, because we do not routinely perform *Aspergillus* spp. PCR on respiratory samples. The CAPA criteria by White identified seven putative CAPA cases; as stated for the Gangneux algorithm, the diagnostic yield of this criteria is low in our cohort because we do not routinely perform *Aspergillus* spp. PCR on respiratory specimens and serum (1,3)-β-D-glucan.

## 4. Discussion

In our retrospective observational study on COVID-19 critically ill mechanically ventilated patients admitted to ICU, using the consensus definition proposed by ECMM/ISHAM, probable CAPA was diagnosed in 11.9% of patients, with a crude incidence rate of 8 cases per 1000 patient days (95% CI 5.1–12.4). Following post-mortem lung examination performed in 6 out of 13 cases (46.2%), proven CAPA was confirmed in four cases, while *Aspergillus* spp. PCR was positive in three cases. The overall autoptic prevalence of CAPA during the study period was 4.8 % (6 out of 125 autopsies).

Incidence of CAPA diagnosed in vivo varies between different studies, especially in monocentric observational studies, where it ranges from 1.1 to 47.4% in mechanically ventilated patients [27,29]. In our study the incidence was similar to data reported in larger multicentric observational studies, where it is around 10–15% [13,15,30,31], and in two systematic reviews, where the rate was 10.2% and 13.5%, respectively [32,33]. Different study designs, awareness of the disease, availability of diagnostic tests and the use of different diagnostic algorithms/case definitions may account for the variation of CAPA incidence in different studies. Mortality in our cohort was extremely high (65%), but consistent with previous reports and data from the two systematic reviews [13,15,32,33].

The prevalence of proven invasive mould disease (IMD) on autopsies is probably lower than incidence rates reported by observational studies, but the estimates are not always concordant. Since the publication of one of the first proven CAPA cases in 2020 [34], a number of studies were then published. In a systematic review of autopsies performed not later than September 2020, the autoptic prevalence of IMD was 2% and *Aspergillus* spp. was the agent in eight cases [19]. Egger and colleagues correctly note some limitations of this systematic review: fungal staining of lung tissue was not routinely performed, ventilated patients were only 58% and the number of patients receiving corticosteroids or anti-interleukin 6 drugs, which have been found in some studies to be associated with CAPA, was <10% [35]. However, we must note that standard autopsy was the most-used technique (81.2% of the decedents and all IMD cases) and this should have ensured an adequate tissue sampling [19]. A study by Flikweert and colleagues in the Netherlands published early in 2020 highlights the issue of lung sampling. The Authors report the results of CT or ultrasound-guided core needle biopsies performed on both lungs of six patients with probable CAPA, whose clinical characteristics (mechanical ventilation, positive GM on BAL) are similar to our cohort [36]. Surprisingly, none of the biopsy specimens showed signs of aspergillosis, but this study has two major limitations: (1) complete autopsy was not performed, thus, even if the biopsy was guided by imaging, lung sampling is not sufficient; (2) *Aspergillus* spp.-specific PCR was not performed. In an Italian study of 45 consecutive complete autopsies, proven CAPA was diagnosed in 20% (nine patients) [37]. This cohort differs substantially from our cases: fungal biomarkers were tested in only four patients (and turned positive in two of them—serum BDG and GM on tracheal aspirate) and antifungal treatment was initiated in only two cases. The clinical suspicion of CAPA was certainly low and autoptic results were probably unexpected. Of note, corticosteroid use was common (88%) and similar to our cohort [37]. In our study, standard autopsies were performed, thus lung samples were adequate.

An important issue to address is whether CAPA is a major driver of mortality on his own, or if it is a collateral finding in COVID-19 critically ill patients whose clinical status is already and irreversibly compromised by viral ARDS. In the previously cited autopsy series, angioinvasion was not a universal finding (5/9 cases in the series by Fortarezza and 2/11 autopsies in the systematic review by Kula) and the main histological pattern seems to be localized fungal pneumonia on a lung background of diffuse alveolar damage and vascular injuries [19,23,37]. This may also explain why, in the study by Flikweert and colleagues, lung biopsies did not show any sign of aspergillosis [36]. In our study, fungal lesions were focal and there were no signs of angioinvasion. Establishing the cause of death in these patients is challenging, but we must note that the predominant lung injury pattern seems to be viral ARDS rather than fungal pneumonia.

Another unresolved question is whether IPA in COVID-19 patients behaves in the same way as in neutropenic patients. IPA in the neutropenic is characterized by marked angioinvasion, which is frequently accompanied by a positive serum GM [38]. The sensitivity of serum GM is lower in non-neutropenic patients, such as in COVID-19, and this biomarker is often negative, especially early in the natural history of the disease [38,39]. This probably reflects a different lung pattern of the disease, initially characterized by a greater airway extension instead of angioinvasion, which determines high positivity rates of BAL GM. As the disease progresses, *Aspergillus* hyphae may be able to invade the lung epithelium and the bloodstream; the frequency of positive serum GM thus increases and this is associated with poor outcome, as reported in a recent study [40]. In our cohort, all patients with a positive serum GM (who also had a positive GM on BAL) died despite treatment. In all but one, serum GM was positive the day of probable CAPA diagnosis or immediately after. In patient #10, serum GM became positive at the 16 days cut-off after CAPA diagnosis.

The uncertainty about the real clinical burden of CAPA is, at the same time, the consequence and the effect of the intrinsic limitations of diagnostic tests, algorithms, and case definitions. Even if there are not conclusive data about the diagnostic performance of tests for CAPA, sensitivity is generally low, as widely reported in many studies and reviews, with good specificity [22,38,39,40,41]. Respiratory samples usually have greater positivity rates, in particular BAL GM, which is the most sensitive biomarker, followed by BAL culture, GM Lateral Flow Assay (LFA) on BAL, and *Aspergillus* PCR on BAL. Serum biomarkers, especially GM, are often negative, reflecting little/no fungal angioinvasion [13,14,15,16,40]. In our cohort of patients, in vivo CAPA diagnosis was mainly achieved using GM on BAL samples. The positivity rate of diagnostic tests is consistent with previous reports, except for respiratory specimens’ culture, which was positive in fewer cases (6/20, 30%, of which four BAL samples). False negative cultures may result from suboptimal lung sampling during bronchoscopy because of the high clinical severity of patients and a disease with a focal pattern.

Early during the pandemic, case definitions for CAPA were not available. Authors frequently adapted IAPA diagnostic algorithms to COVID-19 patients, but this approach has some limitations which have been extensively addressed elsewhere [42]. The consensus definition proposed by ECMM/ISHAM had the ambition to make the diagnosis and treatment of CAPA uniform [22]. Although it effectively standardized CAPA diagnosis across studies, this case definition, as well as the others, is far from being accurate. A recent systematic review and meta-analysis showed that 34% of patients reported to have CAPA did not fulfil any research definition [43]. The ECMM/ISHAM consensus definition allows the diagnosis of CAPA with a certain level of probability (possible or probable) with a single positive diagnostic test, either a respiratory specimen culture or positive fungal biomarker on serum/respiratory specimen, but this is not sufficient to distinguish *Aspergillus* spp. colonization from infection [44]. In our case series, some patients with a probable CAPA diagnosis recovered without antifungal treatment (#13 and #16) and, in case #6 and #7, CAPA was not proven even if a clinical diagnosis was established in vivo.

Considering all issues related to CAPA diagnosis, the clinical picture that emerges could be the following. Among critically ill COVID-19 patients admitted to ICU, a subset of patients has one or more positive diagnostic fungal tests and thus fulfils criteria for probable CAPA. Positive and negative predictive values (PPV and NPV) of the tests may help clinicians in distinguishing invasive infection from colonization. Assuming a CAPA incidence of 10% (close to our findings) and using BAL GM as a diagnostic test with a sensitivity of 75–80% and specificity of 90–94% (which are closer to the performance in COVID-19 patients, as reported by Egger and colleagues) [41], PPV ranges from 45 to 60% and NPV is close to 100%, thus anticipated CAPA incidence sits around 3–5% [45]. This rate is consistent with proven CAPA incidence in autoptic studies [19]. These calculations are dynamic and can be tailored to local epidemiology and diagnostic test performance. Giving the suboptimal PPV of the test, in order to increase diagnostic certainty, efforts should be made in selecting high-risk patients to test (i.e., longer ICU stay, receipt of anti-interleukin-6 drugs, clinical deterioration not explained by other causes, underlying host risk factors) and treatment initiation should be decided on a personal basis, according to clinical, radiological and laboratory parameters [45]. The high NPV suggests that empirical antifungal treatment is not usually warranted in severe COVID-19 patients.

The main strength of our study is the availability of autoptic studies which help the interpretation of clinical data. Histopathological examination of lung tissues is the gold standard for the diagnosis of pulmonary aspergillosis. We also discuss the performance of different CAPA case definitions in a setting with high awareness of the disease, where patients are systematically screened for fungal infections and all the mycological data are reviewed by an expert ID physician. Moreover, even if the timeframe of the study is long (ten months), the risk of death during different epidemic waves did not change significantly, thus the a priori probability of developing CAPA should be constant [21].

Our study has, however, several limitations. First, its monocentric and retrospective design limits the generalizability of the data reported. Second, autopsies were not available for all CAPA patients, thus incidence of proven CAPA is not accurate. Third, our cohort does not include patients from the first epidemic wave, which were excluded because the high and unexpected pressure on our health facility limited our ability to set up a systematic screening for pulmonary aspergillosis at the beginning of the pandemic. Fourth, as we do not routinely perform *Aspergillus*-specific PCR on serum or respiratory samples, neither serum (1,3)-β-D-glucan nor *Aspergillus* rapid diagnostic tests, our diagnostic armamentarium is somewhat limited.

## 5. Conclusions

In this retrospective observational study of mechanically ventilated COVID-19 patients, we found that incidence and mortality of CAPA diagnosed with case definitions are consistent with data from the literature. A high number of complete autopsies was performed, and CAPA was proven in 66.7% of cases. The pattern of IPA and the extent of lung damage secondary to SARS-CoV-2 infection suggests that the main driver of mortality could be viral ARDS rather than IPA. Diagnosing CAPA is challenging because of the weaknesses of diagnostic tests and case definitions. The epidemiological context, host risk factors, as well as clinical, radiological and laboratory parameters should be carefully considered when deciding whom to test and whom to treat. Further studies based on histopathological findings are needed to provide accurate estimates of CAPA incidence and, consequently, to establish optimal diagnostic and treatment strategies.

## Figures and Tables

**Figure 1 jof-08-00894-f001:**
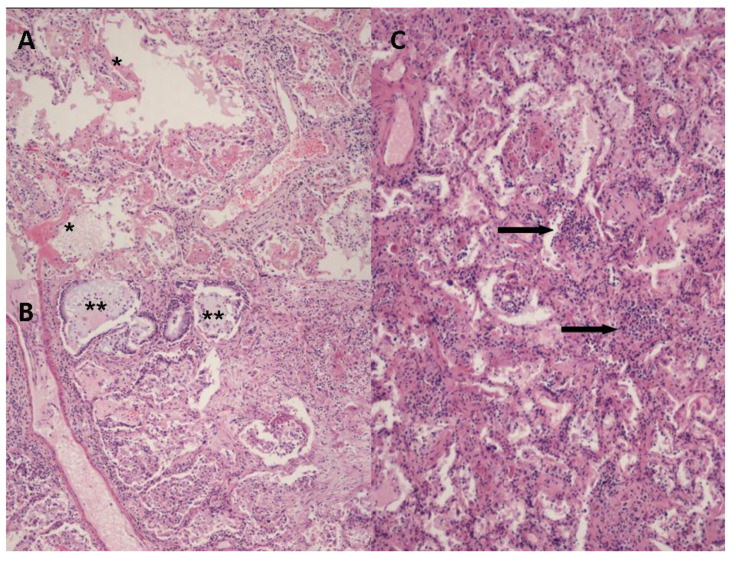
Hematoxylin and eosin-stained sections from representative areas of lung parenchyma with diffuse alveolar damage, characterized by plurifocal hyaline membranes (panel **A**, *), prominent type II pneumocyte hyperplasia and atypia (panel **B**, **) and chronic inflammatory infiltrate (lymphocytes and macrophages) (Panel **C**, black arrows). OM panels (**A**) and (**B**) 10×, panel (**C**) 20×.

**Figure 2 jof-08-00894-f002:**
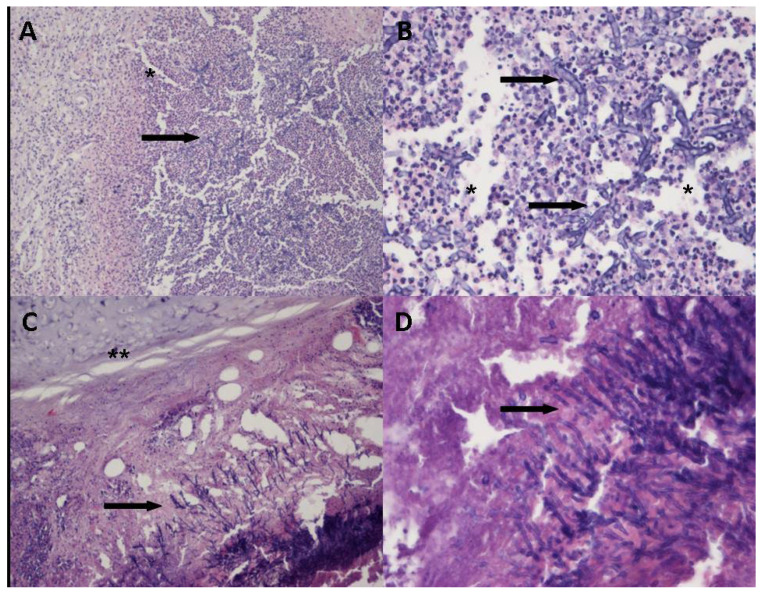
Hematoxylin and eosin-stained sections from large necrotic area of the lung with exudative inflammatory and necrotic lesions (Panels **A** and **B**, *) and acute inflammation and necrosis of the bronchial wall (panels **C** and **D**, **); numerous typical fungal hyphae of *Aspergillus* are present (black arrows). OM panels (**A**) and (**C**) 10×, panels (**B**) and (**D**) 40×.

**Table 1 jof-08-00894-t001:** Characteristics of the study population.

			Overall n = 168	With CAPA n = 20	Without CAPA n = 148	*p* Value
Age, median [IQR]			66 (59–72)	66 (60–72)	66 (59–72)	0.872
Biological sex (%)		Female	39 (23.2)	7 (35)	32 (21.6)	0.256
		Male	129 (76.8)	13 (65)	116 (78.4)	
Obesity *, *n* (%)			60 (35.7)	8 (40)	52 (35.1)	0.804
Respiratory disorders, n (%)			17 (10.1)	3 (15)	14 (9.5)	0.431
Cardiovascular disorders, n (%)			96 (57.1)	9 (45)	87 (58.8)	0.336
Diabetes, n (%)			29 (17.3)	3 (15)	26 (17.6)	0.999
Chronic kidney disease, n (%)			7 (4.2)	1 (5)	6 (4.1)	0.595
Oncological disorders, n (%)			14 (8.3)	3 (15)	11 (7.4)	0.222
Immune system disorders, n (%)			7 (4.2)	3 (15)	4 (2.7)	*0.037*
Hepatic disorders, n (%)			4 (2.4)	0 (0)	4 (2.7)	0.999
Number of comorbidities, n (%)		0	50 (29.8)	7 (35)	43 (29.1)	0.229
		1	51 (30.4)	6 (30)	45 (30.4)	
		2	50 (29.8)	3 (15)	47 (31.8)	
		3+	17 (10.1)	4 (20)	13 (8.8)	
Days from symptom onset to ICU, median [IQR]			10 (7–14)	10 (7–14)	10 (6–14)	0.669
Days from hospital admission to ICU, median [IQR]			2 (0–5)	3 (1–5)	1.5 (0–5)	0.17
Days from ICU admission to MV, median [IQR]			0 (0–1)	0 (0–1)	0 (0–1)	0.845
Days of mechanical ventilation, median [IQR]			11 (6–22)	17 (9–25)	10 (5–21)	0.061
SOFA score, median [IQR]			9 (7–10)	10 (8–11)	9 (7–10)	0.097

Abbreviations. n, number; IQR, Inter Quartile Range; CAPA, COVID-19-associated pulmonary aspergillosis. * Obesity is defined as a body mass index ≥ 30.

**Table 2 jof-08-00894-t002:** Characteristics of patients with a CAPA diagnosis.

	Sex	Age	Comorbidities	Days from ICU Admission to CAPA Diagnosis	BAS/BAL Culture	S-GM Ag Index	BAL-GM Ag Index	Radiological Findings	Concomitant VAP/VAT Microbiology	CAPA Treatment	Outcome	Proven CAPA by Autopsy
#1	M	64	Dyslipidemia	12	Negative (BAL)	<0.5	1	Chest X ray showing progression of the interstitial-alveolar involvement	VAP *S. marcescens*	Voriconazole	Death	Yes
#2	M	72	Hypertension, obesity, diabetes	19	Negative (BAL)	<0.5	1.2	Chest X ray showing progression of alveolar involvement	VAP *P. aeruginosa*	Voriconazole	Recovery	Not performed
#3	F	56	Hypertension, diabetes, previous leukemia and breast cancer	16	Few mould colonies (BAL)	<0.5	Not performed	Chest X ray showing progression of the alveolar involvement	VAP *E. coli*	None/palliative care	Death	Not performed
#4	M	53	Hypothyroidism	9	Negative (BAL)	<0.5	7.9–10.2	Chest X ray unchanged	VAT *K. pneumoniae*	Voriconazole	Recovery	Not performed
#5	M	76	Hypertension, dyslipidemia, ischemic heart disease, diabetes, chronic kidney disease	13	Rare colonies of *A. fumigatus* (BAL)	0.7	7.3–13.5	Chest X ray showing progression of the alveolar involvement	VAP *S. marcescens*	Voriconazole	Death	Not performed
#6	F	73	Obesity, asthma, obstructive sleep apnea, hypothyroidism	0	*A. niger* (BAS)	<0.5	5.1–0.5	CT scan showing ground glass and non-specific interstitial thickening		Voriconazole	Death	No
#7	M	65	None	8	Negative (BAL)	<0.5	6.9	Chest X ray showing progression of the interstitial-alveolar involvement	VAP *P. aeruginosa* and *K. pneumoniae*	Voriconazole	Death	No
#8	F	76	Hypertension, total hip arthroplasty	6	Negative (BAL)	Not performed	8	Chest X ray showing improvement	VAT *S. marcescens*	Voriconazole	Death	Not performed
#9	M	68	Chronic obstructive pulmonary disease, hypothyroidism	2	Negative (BAL)	1.7	12.5	Chest X ray showing progression of the interstitial-alveolar involvement and pneumothorax		Voriconazole	Death	Yes
#10	M	61	Hypertension, diabetes, hypothyroidism	1	Negative (BAL)	0.5	14–1.2	Chest X ray unchanged		Voriconazole	Death	Not performed
#11	M	67	Previous cholecistectomy	5	*A. fumigatus* (BAL)	<0.5	<0.5	CT scan showing ground glass and non-specific interstitial thickening		Voriconazole	Death	Not performed
#12	M	72	Ischemic heart disease, pulmonary disease	10	Negative (BAL)	<0.5	11.5	CT scan showing ground glass and non-specific interstitial thickening		None/palliative care	Death	Yes
#13	M	52	None	6	Negative (BAL)	<0.5	8.3	Chest X ray unchanged		None	Recovery	Not performed
#14	M	66	Ischemic heart disease, dyslipidema	9	Negative (BAL)	<0.5	1.4–0.9	Chest X ray showing progression of the alveolar involvement and new consolidations		Voriconazole	Recovery	Not performed
#15	M	72	None	7	Negative (BAL)	<0.5	1	Chest X ray showing progression of the alveolar involvement		None/palliative care	Death	Not performed
#16	F	55	Pulmonary disease	9	Negative (BAL)	<0.5	1.4	Chest X ray unchanged		None	Recovery	Not performed
#17	F	73	Obesity, hypertension, previous breast cancer	1	Rare colonies of *A. fumigatus* (BAS)	<0.5	10.6	Chest X ray showing improvement		Isavuconazole	Recovery	Not performed
#18	M	45	None	1	Negative (BAL)	<0.5	11.12	Chest X ray showing progression of interstitial-alveolar involvement and new consolidation		Voriconazole	Recovery	Not performed
#19	F	71	Previous hematological malignancy	10	Rare colonies of *A. fumigatus* (BAL)	1.0–1.9	2.2	Chest X-ray unchanged	VAP *E. aerogenes*	Voriconazole	Death	Not performed
#20	F	67	Capillary leak syndrome, previous breast cancer	10	Negative (BAL)	0.5	4.7	Chest X-ray showing new consolidation	VAP *K. pneumoniae*	None	Death	Yes

Abbreviations. ICU, intensive care unit; CAPA, COVID-19-associated pulmonary aspergillosis; BAS, bronchial aspirate; BAL, bronchoalveolar lavage; GM, galactomannan; S, serum; Ag, antigen; VAP, ventilator-associated pneumonia; VAT, ventilator-associated tracheobronchitis; CT, computed tomography.

**Table 3 jof-08-00894-t003:** Categorization of patients with CAPA according to the different proposed case definitions and algorithms.

	IAPA/CAPA Verweij et al.	*Asp*ICU Blot et al.	Modified *Asp*ICU Gangneux et al.	CAPA White et al.	ECMM/ISHAM Koehler et al.	IAPA/CAPA Schauwvlieghe et al.	Proven CAPA by Autopsy
#1	Probable	NC	NC	NC	Probable	Probable	Yes
#2	Probable	NC	NC	NC	Probable	Probable	Not performed
#3	Probable	Colonization	Colonization	NC	Probable	Probable	Not performed
#4	Probable	NC	NC	Putative	Probable	Probable	Not performed
#5	Probable	Colonization	Colonization	Putative	Probable	Probable	Not performed
#6	Probable	Colonization	Colonization	NC	Probable	Probable	No
#7	Probable	NC	NC	NC	Probable	Probable	No
#8	Probable	NC	NC	NC	Probable	Probable	Not performed
#9	Probable	NC	NC	Putative	Probable	Probable	Yes
#10	Probable	NC	NC	Putative	Probable	Probable	Not performed
#11	Probable	Colonization	Colonization	NC	Probable	Probable	Not performed
#12	Probable	NC	NC	NC	Probable	Probable	Yes
#13	Probable	NC	NC	NC	Probable	Probable	Not performed
#14	Probable	NC	NC	NC	Probable	Probable	Not performed
#15	Probable	NC	NC	NC	Probable	Probable	Not performed
#16	Probable	NC	NC	NC	Probable	Probable	Not performed
#17	Probable	Colonization	Colonization	Putative	Probable	Probable	Not performed
#18	Probable	NC	NC	NC	Probable	Probable	Not performed
#19	Probable	Colonization	Colonization	Putative	Probable	Probable	Not performed
#20	Probable	NC	NC	Putative	Probable	Probable	Yes

The categorization was performed using in vivo clinical data; patients with autopsy-confirmed CAPA would have been classified as “Proven CAPA” with all case definitions and algorithms. Abbreviations: NC, not classifiable; CAPA, COVID-19-associated pulmonary aspergillosis; IAPA, Influenza-associated pulmonary aspergillosis; ICU, Intensive care unit.

## Data Availability

Data will be made available by the corresponding author on request as far as data protection of the patients can be warranted.

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
