# Peer review of "Challenges in Diagnosing COVID-19-Associated Pulmonary Aspergillosis in Critically Ill Patients: The Relationship between Case Definitions and Autoptic Data"

_jof, 2022, doi:10.3390/jof8090894_

Round 1

Reviewer 1 Report

In this manuscript authors describe a monocentric experience with CAPA patients, where they describe how challenging is the timely diagnosis of such cases and that definition-based diagnosed might not be always in line with what observed in the ulterior autopsies.

It is well written and it is easy to read. However, I would like to stress some aspects which I consider should be reviewed:

- I would suggest to rearrange the Introduction in 3 paragraphs: 1) What do we know?, 2) What do we not know?, and 3) Which are our aims (the current last paragraph).

- The patients were screened between August 2020 and May 2021. Currently, we are in August 2022. I was wondering whether it could be possible to collect further patients and provide data with more updated details.

- Regarding antifungal treatment, I was wondering whether it could be possible to include, at least in table 2, a further explanation about it: dosage, length, formulation, TDM...

- Could it be possible to provide the number of follow up days in the "Outcome" column of table 2? Was there any difference with this regards between CAPA and non-CAPA patients?

- Regarding patient #3, Aspergillus is not considered normally as a "rare mould", I was wondering whether you could explain or clarify why a patient without BAL GM and a negative serum GM was considered to have CAPA with "rare mould colonies in BAL" and not, for example COVID-19 associated mucormycosis.

- Could it be possible to provide details on the time from MV intiation and CAPA diagnosis?

- In table 3, I would suggest to mention that the classification in the definitions from Verweij to Schauwvlieghe you are considering the in-vivo classification, as the patients with a positive autopsy would be proven CAPA in an overall classification.

- Several publications have provided in the past (Salmanton-García et al. EID, García-Vidal et al. EID, etc) data on the incidence of CAPA in different centres around the would. I would suggest to include such numbers in your introduction and or discussion, so you can compare your numbers with other monocentric experiences.

Author Response

Response to reviewer #1

In this manuscript authors describe a monocentric experience with CAPA patients, where they describe how challenging is the timely diagnosis of such cases and that definition-based diagnosed might not be always in line with what observed in the ulterior autopsies.

It is well written and it is easy to read. However, I would like to stress some aspects which I consider should be reviewed:

- I would suggest to rearrange the Introduction in 3 paragraphs: 1) What do we know?, 2) What do we not know?, and 3) Which are our aims (the current last paragraph).

We thank the reviewer for the useful suggestion. Introduction was revised according to suggestions.

- The patients were screened between August 2020 and May 2021. Currently, we are in August 2022. I was wondering whether it could be possible to collect further patients and provide data with more updated details.

We thank the reviewer for this comment, and we agree with the reviewer on the fact that the period of observation could be prolonged. However, the systematic data collection about secondary fungal infections in mechanically ventilated COVID-19 patients admitted to ICU was not consistent after May 2021. Furthermore, autopsies were performed on a regularly basis only during the study period, thus a complete subset of data is available only for the study period that we chose.

- Regarding antifungal treatment, I was wondering whether it could be possible to include, at least in table 2, a further explanation about it: dosage, length, formulation, TDM...

We thank the reviewer for this useful suggestion. Details about antifungal treatment have been included in the “Result” section.

- Could it be possible to provide the number of follow up days in the "Outcome" column of table 2? Was there any difference with this regards between CAPA and non-CAPA patients?

We are grateful to the reviewer for this comment. Patients’ outcome refers to hospital status (alive at home discharge – recovery, or hospital death). Considering that for statistical purposes the follow-up of patients ends with the “CAPA event”, it could be more useful to compare the median time of mechanical ventilation between the two groups of patients. In patients with probable CAPA it was 17 days (IQR 9-25), while patients without CAPA received mechanical ventilation for a median of 10 days (IQR 5-21). This result is now on Table 2.

- Regarding patient #3, Aspergillus is not considered normally as a "rare mould", I was wondering whether you could explain or clarify why a patient without BAL GM and a negative serum GM was considered to have CAPA with "rare mould colonies in BAL" and not, for example COVID-19 associated mucormycosis.

We thank the reviewer for this comment. With “rare mould colonies” we intend that the number of grown colonies was low, and not that it was a “rare mould”. We have corrected the table. Although the mould was not identified with certainty, considering the local epidemiology and the cases of CAPA observed previously, we assumed that the observed colonies were from Aspergillus spp.

- Could it be possible to provide details on the time from MV initiation and CAPA diagnosis?

We thank the reviewer for this question. The median time from orotracheal intubation and CAPA diagnosis was 8.5 days (IQR 4-10), which is very close to the median time from ICU admission to CAPA diagnosis because most patients were intubated on ICU admission or immediately after, as stated in the paper. We included this result in the paper.

- In table 3, I would suggest to mention that the classification in the definitions from Verweij to Schauwvlieghe you are considering the in-vivo classification, as the patients with a positive autopsy would be proven CAPA in an overall classification.

We thank the reviewer for this useful suggestion. We have mentioned it in the description of Table 3.

- Several publications have provided in the past (Salmanton-García et al. EID, García-Vidal et al. EID, etc) data on the incidence of CAPA in different centres around the world. I would suggest to include such numbers in your introduction and or discussion, so you can compare your numbers with other monocentric experiences.

We thank the reviewer for this comment. We included one of the references suggested to further expand the discussion.

Reviewer 2 Report

This study uses a limited number of autopsies to try to prove or disprove various case definitions for CAPA that are based on laboratory criteria.  While it uses autopsy in the title, only 6/20 of their probably CAPA patients actually had an autopsy.  The authors did a very good job of explaining the rest of the limitations of this manuscript in their discussion.

When you have histopathology slides, why was a silver stain not performed.  This would have been the gold standard for fungi, not H&E.  This seems to be a huge oversight.

Line 351: Consider that these patients may have recovered underlying immune function and therefore overcame their infection rather than being simply colonized the whole time.

Author Response

Response to reviewer #2

This study uses a limited number of autopsies to try to prove or disprove various case definitions for CAPA that are based on laboratory criteria.  While it uses autopsy in the title, only 6/20 of their probably CAPA patients actually had an autopsy.  The authors did a very good job of explaining the rest of the limitations of this manuscript in their discussion.

When you have histopathology slides, why was a silver stain not performed.  This would have been the gold standard for fungi, not H&E.  This seems to be a huge oversight.

We thank the reviewer for this comment, and we agree that methenamine silver staining (Grocott-Gomori)  is the best stain for detection of all fungi and it was also performed by our pathologists. However, we chose to use a picture stained with H&E in order to show the host reaction in infected tissue. Finally, we performed Aspergillus PCR on tissue samples to further confirm the diagnosis of aspergillosis.

Line 351: Consider that these patients may have recovered underlying immune function and therefore overcame their infection rather than being simply colonized the whole time.

We thank the reviewer for this comment, and we agree with him/her that immune reconstitution may contribute to reduce the risk of developing overt invasive aspergillosis. However, we have no clinical or laboratory data to support this hypothesis and we can simply state that diagnostic algorithms are not always sufficient to distinguish Aspergillus spp. colonization from infection.
